# Association of membership in a farmer producer organization with crop diversity, household income, diet diversity, and women's empowerment in Uttar Pradesh, India

**Lindsay M. Jaacks**[1], **Nishmeet Singh**[1], **Divya Veluguri**[1], **Kaela Connors**[1], **Aleesha Sooraj**[2], **Apoorva Kalra**[2], **Ananya Awasthi**[2]*

1 Global Academy of Agriculture and Food Systems, The University of Edinburgh, Midlothian, UK,
2 Anuvaad Solutions, New Delhi, India

* awasthi@anuvaad.org.in

## Abstract

### Introduction

To date, the impact of farmer producer organizations (FPOs) in India is under-explored in the empirical literature. The primary objective of this study was to test whether agricultural households with FPO members in Uttar Pradesh, India have greater crop diversity and higher income, and whether adults in those households have greater diet diversity and women are more empowered.

### Methods

A cross-sectional survey was conducted in 2023 with two groups of agricultural households, those: (1) with an FPO member (n = 417 households, 414 men and 412 women) and (2) without an FPO member (n = 402, 395 men and 388 women). Diet diversity was measured at the individual level using the FAO minimum diet diversity (MDD) score. Women's empowerment was measured using a modified version of the Abbreviated Women's Empowerment in Agriculture Index (A-WEAI) score. The association between FPO membership and crop diversity, household income, MDD, and A-WEAI was estimated using separate regressions, adjusted for educational attainment, caste, farm size, and, for household income, number of household members.

### Results

FPO households had slightly greater crop diversity than non-FPO households (mean of 4 crops versus 3.5 crops, respectively). FPO households were more likely to have an income from cultivation and livestock than non-FPO households. Hence, FPO households had higher total annual household income than non-FPO households. Adults in FPO households were more likely to have diverse diets. However, they also had greater consumption

**Data availability statement:** The anonymized dataset necessary to replicate our study findings is available on the Harvard Dataverse at: https://doi.org/10.7910/DVN/QWQRS7 Code files necessary to replicate our study findings are available on GitHub at: https://github.com/snishmeet1699/FPO-Evaluation-India

**Funding:** Bill and Melinda Gates Foundation.

**Competing interests:** The authors have declared that no competing interests exist.

of unhealthy foods. There was not a significant difference in women's empowerment between FPO and non-FPO women.

## Conclusion

These findings suggest that FPOs are a potentially effective strategy for improving farmers' incomes, and that this has mixed effects on the healthfulness of household diets. This information can be used to inform evidence-based policies to provide dedicated support for promotion of FPOs and for improved convergence between rural development, agriculture, and nutrition.

## Introduction

Farmer cooperatives have been recognized as one approach to increasing the income of agricultural households. Cooperatives already cover a substantial proportion of the agricultural sector in many developed economies. For example, in Austria, Denmark, Finland, France, Ireland, the Netherlands, and Sweden, cooperatives' agricultural market share exceeds 50% [1]. Many studies have demonstrated that membership in farmer cooperatives is associated with an increase in household income, though findings are mixed as to whether farms of all sizes benefit equally [2,3].

Few, if any, studies have explored the potential impact farmer cooperatives may have on other important outcomes for agricultural communities, such as dietary intake and women's empowerment [4]. Given that agricultural households around the world purchase a majority of the food they consume [5], any impact cooperatives have on income could in turn influence diets. Farmer cooperatives could also influence diets through improving yields, though the evidence is mixed on whether cooperative membership has a significant effect on yields [6]. Finally, farmer cooperatives could influence diets by changing the types of crops cultivated and increasing crop diversity [7], though these outcomes are relatively unexplored in the literature.

The democratic decision-making process characteristic of many farmer cooperatives may boost women's empowerment. A study in Uganda found that being part of a cooperative increased women's negotiating skills and decision-making power [8]. A study in Ethiopia found that cooperative membership increased the market price and quantity sold by women honey farmers [9].

Agriculture employs 44% of India's population [10] and the vast majority of farmers are smallholders [11]. Farmer producer organizations (FPOs) have been proposed as a solution to stagnating yields [12] in the face of rising cultivation costs [13]. In 2020, the Government of India launched a new scheme called the "Formation and Promotion of 10,000 Farmer Producer Organisations (FPOs)" [14]. According to the latest data from the Ministry of Corporate Affairs, 26,938 FPOs are classified as active and compliant with all necessary regulatory filings and requirements, though only 15,455 of those FPOs have submitted their financials as of 2023 [15]. In this context, the impact of FPOs is under-explored in the empirical literature [16–21]. A few studies have found FPO membership to be associated with higher incomes [17,19]. For example, a study comparing 274 FPO households to 276 non-FPO households in Bihar found that, on average, FPO household income was ~ Rs.2,200/month more than non-FPO households [17]. Likewise, a study comparing 200 FPO households to 100 non-FPO households in Gujarat found that, on average, FPO household income was ~ Rs.778/month more than non-FPO households [19].

The state of Uttar Pradesh in northern India has been at the forefront of setting up FPOs with a dedicated policy, "Uttar Pradesh Farmer Production Organization Policy, 2020" [22]. A state-level project management unit has been set up to support the formation, promotion, and strengthening of FPOs, acting as a resource center for FPOs across the state. The primary objective of this study was to test whether agricultural households with FPO members in Uttar Pradesh, India have greater crop diversity and higher cultivation income, and whether adults in those households have greater diet diversity and women are more empowered. Secondary objectives were (1) to assess the market availability of nutrient-dense produce in villages with and without FPO members and (2) to understand existing FPO support services and perceived challenges to founding and operating FPOs in this context. Findings are expected to inform evidence-based policies to provide dedicated support for promotion of FPOs by the government and implementation partners.

## Methods

### Study context

The study took place in Fatehpur district of Uttar Pradesh. According to the latest National Family Health Survey (2019–2021), 75% of households in Uttar Pradesh are in rural areas and only 66% of women aged 15–49 years are literate, compared to 82% of men and a national average for women of 72% [23]. Uttar Pradesh experiences a significant double burden of malnutrition. While half of women have anemia and one-fifth of men, more than half of women (57%) and men (52%) have a high waist-to-hip ratio that puts them at risk of metabolic diseases such as type 2 diabetes [23].

With respect to agricultural production in India (2021–2022), Uttar Pradesh is first in food grains production, contributing 18% of the country's total production [24]. While Uttar Pradesh stands second in the production of rice (12% of the country's total production) and first in wheat (32%), it is fifth in the production of coarse cereals (8%) and pulses (9%) [24]. The cost of production in Uttar Pradesh for paddy and wheat (2021–2022) is Rs.1,287/Quintal and Rs.1,082/Quintal, respectively, and the average monthly income for agricultural households (2018–2019) is Rs.8,061 [24].

An FPO in Fatehpur district was chosen based on its location – relatively near to Lucknow, the state capital, to facilitate data collection – the fact that about 100 of its 750 members were women, and strong support for the study from the FPO's Chief Executive Officer (CEO). The FPO evaluated was formed out of an existing farmer interest group with support from the National Bank for Agriculture and Rural Development in 2016. At the time of the evaluation, the FPO had a 5-member director board. One of the board members was a woman, and the caste of one board member was other backward caste (OBC). Many women members of the FPO were members of self-help groups in their communities, and their membership in these self-help groups played a role in their joining the FPO.

### Study design

This study employed a mixed methods approach. Quantitative household-level surveys were conducted across 27 villages in the district: 22 villages where FPO members lived and 5 nearby villages without FPO members (herein "control villages"). Nearby villages were purposefully selected to control for potential confounding factors such as weather. Additionally, a market basket survey was conducted in 13 villages: 8 FPO villages and the same 5 control villages. Semi-structured qualitative interviews were conducted with three relevant stakeholders involved in FPO regulation and management.

## Household survey

Household-level interviews were conducted between 28 April and 30 May 2023 by trained enumerators using standardized procedures. The survey is provided in the Supporting Information (S1 File). In each household, the quantitative survey instrument was administered with an adult male and adult female except for in 29 households where only one adult was available for the survey. In most instances (94%), a husband-wife pair was interviewed. Eligibility criteria were as follows: engaged in agricultural work, 18 years or older, and provided informed consent. All the interviews were captured electronically with tablets.

The quota sampling technique was used for respondent selection, meaning that we interviewed participants in FPO and control villages until a sample size of ~ 400 households was achieved for each group. A sample size of 400 per group was determined in power calculations as the sample size required to achieve a 95% confidence interval (CI) with a width of 0.10. A width of 0.10 is typically considered a narrow – i.e., reliable – CI [25,26].

The quantitative survey instrument included modules on demographics adapted from the National Family Health Survey [23]; household income and expenditures and agricultural production adapted from the situation assessment of agricultural households administered by the National Sample Survey Organization (NSSO) [27]; the Diet Quality Questionnaire [28,29]; a slightly modified version of the Abbreviated Women's Empowerment in Agriculture Index (A-WEAI) [30]; and, in FPO households, interaction with the FPO. The slight modifications to the A-WEAI were made in order to reduce participant burden and specifically included: (1) collecting time use in 30-minute intervals instead of 15-minute intervals and (2) not asking, "To what extent do you feel you can make your own personal decisions regarding [ACTIVITY] if you want(ed) to?" as part of the decision-making module because respondents in pilot testing were unable to distinguish between this question and the question, "How much input did you have in making decisions about [ACTIVITY]?" [30]. Interviews with each respondent took approximately 60 minutes.

## Market basket survey

A market basket survey was conducted on 4 and 5 April 2023 through direct observation of all open and operating vendors in 13 of the 27 villages where participants lived. All 5 control villages were selected. The 8 FPO villages were selected based on distance from the FPO office (within ~ 10 km radius) and availability of shops. Vendors included village markets, stationary stores, mobile street vendors, and pan shops (analogous to convenience stores, selling tobacco products and packaged snacks). The survey captured availability of fruits, vegetables, nuts/seeds, and millets. Data were captured electronically using SurveyToGo.

## Semi-structured qualitative interviews

Qualitative interviews were conducted in July 2023. Stakeholders interviewed included the CEO of the FPO, a representative of the state agriculture department, and a district-level government administrative official. The interviews were conducted face-to-face and in one case a written response was received from the interviewee. A semi-structured interview guide was used to administer the interviews. The interview guide was designed with the aim of understanding state-FPO interactions, policies and schemes supporting FPOs in Uttar Pradesh, challenges faced by FPOs in production, especially of nutrient-dense crops, and the support required to overcome these challenges.

## Ethics

The study protocol was reviewed and approved by the Institutional Review Board of the Centre for Media Studies, New Delhi (reference ID: CMS-IRB/Ag/2023/006). All participants were

given a hard copy of a Participant Information Sheet describing, using clear and simple language, what would happen if they took part in the study, procedures relating to confidentiality and anonymity, participants' rights, benefits and risks, and contact details for further information. A trained enumerator slowly and carefully read the Participant Information Sheet out loud, pausing after each section to allow the participant to ask clarifying questions. Informed verbal consent was obtained from all participants before proceeding with the household survey or qualitative interview. Verbal consent was documented directly in the survey software. Additional information regarding the ethical, cultural, and scientific considerations specific to inclusivity in global research is included in the Supporting Information (S1 Checklist).

## Statistical analysis

Using the household survey data, we reported descriptive statistics as a percentage of households or respondents, comparing FPO and control groups. Self-reported demographic variables included total household members, respondent age, gender, marital status, caste, and education. For agriculture-related analyses, we reported seasonal (Kharif and Rabi) and last 12-month variation between the two groups for land cultivated (in acres), number and type of crops, crop-wise yield, whether the produce was sold (yes/no), how much was sold, the sale price, where produce was sold, and farm-related advisories. The unit of yield was missing for 13 wheat and paddy farmers and erroneously reported as "per week" for 12 wheat and paddy farmers; in these instances, it was imputed as "total" to be consistent with the unit reported by 97% of wheat and 96% of paddy farmers in this sample. We excluded six outliers from the yield analysis and 18 outliers from the sold amount and price of produce analysis who reported values > 3 SD above or below the mean value. Analyses of sale quantity, price and selling point were restricted to those farmers who had sold (fully or partially) their produce at the time of the survey. Farm-size was categorized using the total land cultivated in the kharif season as per NSSO categories: landless (0 hectares), small (>0 to 2.0 hectares), medium (>2.0 to 4.0 hectares), large (>4.0 hectares). For income-related analyses, we reported household total income and sources of income.

We quantified diet diversity at the individual level using the Food and Agriculture Organization's minimum diet diversity (MDD) score which ranges from 0 to 10 [31]. The 10 food groups include: starchy foods (e.g., rice, chapati, roti, millets, potatoes, etc.), pulses and legumes (e.g., daal, chana, etc.), nuts and seeds, dairy (e.g., paneer, curd, tea with milk, etc.), meat (including fish), eggs, green leafy vegetables, vitamin A rich fruits and vegetables (e.g., carrot, mango, papaya, etc.), other fruits, and other vegetables. If a respondent consumed the food group on the previous day, they received a point for that food group. Participants with MDD scores ≥ 5 were classified as having a diverse diet. The Diet Quality Questionnaire also includes intake of unhealthy foods on the previous day [28,29], and the frequency of consumption of these foods was also reported.

We used the five domains of the A-WEAI to identify empowered women in our sample [30]. These domains consist of six indicators: (1) input in productive decisions ("production"); (2) ownership of assets and (3) access to and decisions about credit (together, "resources"); (4) control over use of income ("income"); (5) self-help group membership ("leadership"); and (6) workload ("time balance") [30]. We determined adequacy of each indicator according to those previously defined by the A-WEAI [30]. In order to obtain a composite women's empowerment score, we made modifications to the A-WEAI calculation [32]. We determined whether each woman was empowered versus disempowered based on each of the six indicators belonging to the 5 domains. Then, we assigned equal weights (1/5) to each indicator that corresponded to a domain. However, since the indicators "ownership of assets" and "access to and decisions about credit" both belong to the same domain, they were assigned a weight

of 1/10. We determined whether a woman was empowered or not by summing the weighted adequacies across the six indicators. The lowest possible score is 0 which means a woman is not empowered across all domains and the highest possible score is 1 meaning a woman is empowered in all five domains. Finally, women were considered empowered using a cut-off of 0.6 meaning that they were empowered in at least three of the five domains.

The association between FPO membership and crop diversity was estimated using multi-variable linear regression, adjusting for educational attainment (male and female), caste, and total land cultivated in the kharif season (hectares). The association between FPO member-ship and total household income was estimated using a two-part regression model (binary and log-linear models) [33], adjusting for educational attainment (male and female), caste, total land cultivated in the kharif season (hectares), and number of household members. The asso-ciation between FPO membership and diet diversity was estimated first using multivariable logistic regression with a binary outcome (MDD score ≥ 5) and then with multivariable linear regression (MDD score, ranging from 0 to 10). Both diet diversity models were adjusted for sex, educational attainment, caste, and total land cultivated in the kharif season (hectares), and standard errors were adjusted for clustering within households. The association between FPO membership and women's empowerment was estimated using logistic regression (A-WEAI score ≥ 0.6), adjusting for women's educational attainment, caste, and total land cultivated in the kharif season (hectares). The two-part model for income was run using Stata/SE v 17.0. All other analyses were conducted using RStudio v 2024.04.2 Build 764. Complete case analysis was used to handle missing data (<10% data missing for all variables, see Table 1 in S1 Text).

## Results

A total of 417 FPO households (826 adults, 414 men and 412 women) and 402 control house-holds (783 adults, 395 men and 388 women) were interviewed. On average, households had 5 members (Table 1). Men were, on average, 45–46 years old and women were, on average, 42–43 years old. Members of FPO households had higher educational attainments and were more likely to be OBC households compared to non-FPO households. FPO households also had slightly larger farms compared to non-FPO households (mean 0.79 hectares versus 0.68 hectares, respectively) and were more likely to own 2 or more types of livestock.

### Crop diversity

In the year prior to the survey, FPO households cultivated slightly more crops than non-FPO households: adjusted mean difference (95% CI), 0.30 crops (0.09, 0.50), p = 0.005 (Table 2 in S1 Text). Wheat and paddy were the most commonly cultivated crops for both FPO and non-FPO households (Fig 1). Red chili was the most commonly cultivated vegetable.

### Household income

FPO households were more likely to have an income from cultivation and livestock than non-FPO households (Table 1). Among both FPO and non-FPO households, less than 5% had income from other agricultural activity such as selling fodder, fisheries, etc.; non-agricultural enterprises; or salaried employment (combined as 'Other income' in Table 1). FPO households had a higher total monthly household income than non-FPO households: adjusted mean dif-ference (95% CI), 602 INR/month (127.5, 1,076), p = 0.013 (Table 3 in S1 Text). This difference was driven by higher cultivation income (Fig 2).

FPO households reported a slightly higher yield and sale quantity for paddy and wheat than non-FPO households (Table 2). However, FPO households received a slightly lower sale price for their produce and were more likely to sell to a trader than a village market than

**Table 1. Characteristics of participants, farm size, and household crop diversity and income from a cross-sectional survey comparing households with and without farmer producer organization (FPO) members in Uttar Pradesh, India[1].**

| | Overall n = 1,609 | FPO household n = 826 | Non-FPO household n = 783 |
|---|---|---|---|
| **Household members**, count | 5.4 (1.8) | 5.5 (2.0) | 5.3 (1.5) |
| **Age**, years | 43.8 (11.3) | 43.5 (10.9) | 44.1 (11.7) |
| **Educational attainment** | | | |
| No formal school (including illiterate) | 26% (414) | 26% (214) | 26% (200) |
| Primary school | 24% (392) | 19% (161) | 30% (231) |
| High school | 27% (435) | 28% (232) | 26% (203) |
| Secondary school | 15% (242) | 19% (154) | 11% (88) |
| Graduate and above | 7.8% (126) | 7.9% (65) | 7.8% (61) |
| **Caste** | | | |
| Scheduled Caste/Tribe | 20% (319) | 17% (142) | 23% (177) |
| Other Backward Caste | 59% (943) | 70% (580) | 46% (363) |
| General/Other | 22% (346) | 12% (103) | 31% (243) |
| **Farm size** | | | |
| Landless (0 hectares) | 0.7% (6) | 1.4% (6) | 0% (0) |
| Small (>0 to 2.0 hectares) | 96% (780) | 95% (394) | 97% (386) |
| Medium (>2.0 to 4.0 hectares) | 2.6% (21) | 2.9% (12) | 2.3% (9) |
| Large (>4.0 hectares) | 0.5% (4) | 0.5% (2) | 0.5% (2) |
| **Agricultural land owned**, hectares | 0.74 (0.67) | 0.79 (0.75) | 0.68 (0.58) |
| **Cultivated crops**, count | 3.76 (1.59) | 4.02 (1.70) | 3.49 (1.42) |
| **Types of livestock owned**[2] | | | |
| None | 17% (139) | 12% (52) | 21% (87) |
| 1 | 57% (461) | 57% (236) | 56% (225) |
| 2 or more | 27% (219) | 31% (129) | 23% (90) |
| **Total household income**, INR/month | 3,529.6 (4,010.6) | 4,035.3 (4,451.1) | 2,999.5 (3,415.8) |
| Had income from cultivation | 52% (419) | 56% (233) | 47% (186) |
| Cultivation income, INR/month | 2,333.6 (3,843.5) | 2,782.8 (4,354.6) | 1,862.9 (3,159.6) |
| Had income from livestock | 29% (238) | 37% (153) | 22% (85) |
| Livestock income, INR/month | 277.9 (712.2) | 344.0 (787.6) | 208.5 (616.8) |
| Had income from wages | 49% (399) | 45% (187) | 54% (212) |
| Wage income, INR/month | 765.3 (973.5) | 711.8 (974.5) | 821.3 (970.5) |
| Had income from non-agricultural source | 2.3% (19) | 3.1% (13) | 1.5% (6) |
| Other income, INR/month | 152.8 (915.0) | 196.7 (1,023.3) | 106.8 (784.4) |

[1]Values are mean (SD) or % (n).

[2]Includes owning cows, buffaloes, bulls, other large livestock, chickens, goats, sheep, pigs, or other small livestock.

non-FPO households. The proportion of FPO and non-FPO households selling at government markets known as 'mandis' was relatively low, especially for paddy.

## Diet diversity

Overall, 42% of adults in FPO households had diverse diets compared to 31% of adults in non-FPO households (Table 3). FPO households were more likely to have diverse diets than non-FPO households: adjusted odds ratio (95% CI), 1.35 (1.08, 1.68) (Table 4 in S1 Text). Adjusted mean MDD was higher among both men and women in FPO households than in non-FPO

**Table 2. Paddy and wheat yield, sale status, sale quantity, sale price and selling point from a cross-sectional survey comparing households with and without farmer producer organization (FPO) members in Uttar Pradesh, India[1].**

| | Paddy | | Wheat | |
|---|---|---|---|---|
| | FPO household | Non-FPO household | FPO household | Non-FPO household |
| | **n = 228** | **n = 147** | **n = 380** | **n = 333** |
| **Yield**, kg/ha | 3,311.3 (1,143.7) | 3,258.1 (1,157.2) | 3,130.7 (1,199.0) | 3,037.6 (1,299.1) |
| **Sale status** | n = 231 | n = 147 | n = 381 | n = 335 |
| Not sold | 73% (169) | 79% (116) | 59% (223) | 60% (200) |
| Sold partially | 24% (55) | 21% (31) | 38% (146) | 40% (134) |
| Sold fully | 3.0% (7) | 0% (0) | 3.1% (12) | 0.3% (1) |
| **Among those who sold partially or fully** | n = 56 | n = 31 | n = 141 | n = 132 |
| **Sale quantity**, quintals | 12.1 (9.8) | 11.8 (8.2) | 13.7 (9.2) | 11.8 (8.8) |
| **Sale price**, INR/quintal | 1,490.8 (448.7) | 1,653.9 (481.8) | 2,022.3 (459.6) | 2,141.6 (524.9) |
| **Selling point** | | | | |
| Village market | 67% (39) | 61% (19) | 38% (54) | 61% (80) |
| Government market | 3.4% (2) | 6.5% (2) | 27% (39) | 26% (34) |
| Trader | 64% (37) | 55% (17) | 56% (80) | 31% (41) |

[1]Values are mean (SD) or % (n).

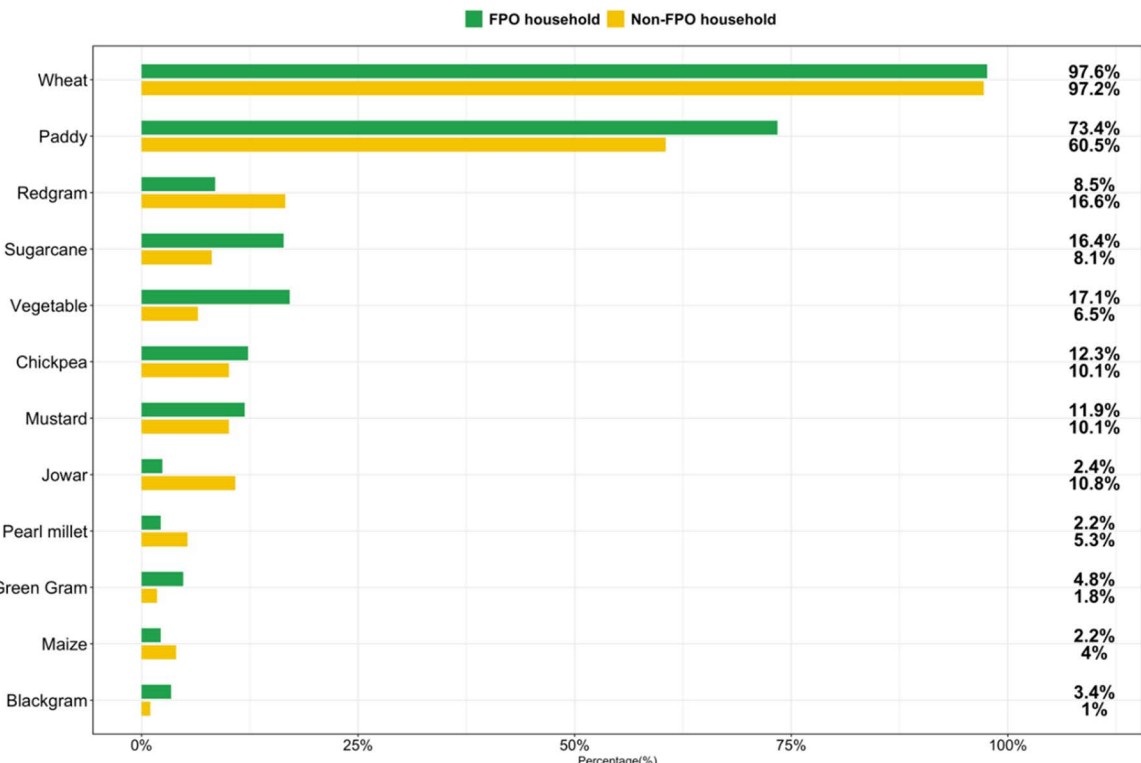

**Fig 1. Crops cultivated by agricultural households participating in a cross-sectional survey comparing households with and without farmer producer organization (FPO) members in Uttar Pradesh, India (n = 809 households).**

**Table 3. Dietary intake of adults participating in a cross-sectional survey comparing households with and without farmer producer organization (FPO) members in Uttar Pradesh, India[1].**

| | Overall n = 1,609 | FPO household n = 826 | Non-FPO household n = 783 |
|---|---|---|---|
| Minimum dietary diversity score | 4.1 (1.3) | 4.3 (1.4) | 4.0 (1.2) |
| Diverse diet (minimum dietary diversity score ≥5) | 36% (586) | 42% (344) | 31% (242) |
| **Nutritious foods** | | | |
| Gourds | 20% (321) | 27% (220) | 13% (101) |
| Cucumber, capsicum, drumstick | 24% (390) | 29% (241) | 19% (149) |
| Green Leafy: mustard, spinach, other | 8.2% (132) | 9.2% (76) | 7.2% (56) |
| Papaya, mango | 20% (319) | 24% (196) | 16% (123) |
| Banana, apple, watermelon | 13% (202) | 18% (150) | 6.6% (52) |
| Grapes, peaches, jackfruit | 15% (241) | 22% (183) | 7.4% (58) |
| Paneer | 1.5% (24) | 2.2% (18) | 0.8% (6) |
| Curd | 31% (500) | 37% (306) | 25% (194) |
| Milk | 55% (890) | 52% (433) | 58% (457) |
| Fish | 35% (184) | 40% (97) | 31% (87) |
| Peanuts, cashews, almonds, pistachios, walnuts, pumpkin seeds, or sunflower seeds | 1.6% (25) | 1.5% (12) | 1.7% (13) |
| **Unhealthy foods** | | | |
| Cake, biscuits, halwa, jalebi, ladoo | 55% (892) | 60% (498) | 50% (394) |
| Other mithai, kulfi, ice cream, shakes | 9.7% (156) | 15% (123) | 4.2% (33) |
| Chips, namkeen | 6.7% (107) | 10% (85) | 2.8% (22) |
| Maggi noodles, wai wai | 8.6% (138) | 12% (103) | 4.5% (35) |
| Samosa, pakora, puri, vada | 44% (701) | 49% (401) | 38% (300) |
| Fruit juice, frooti | 7.5% (121) | 10% (84) | 4.7% (37) |
| Cold drinks | 17% (281) | 23% (189) | 12% (92) |

[1]Values are mean (SD) or % (n).

households (Fig 3, and Table 5 in S1 Text). Greater diet diversity among FPO households compared to non-FPO households was especially evident in those reporting being from a Scheduled Caste/Tribe or OBC rather than General/Other caste (Table 6 in S1 Text).

Vegetables, fruits, dairy, and fish were the main foods contributing to greater diet diversity in FPO households (Table 3). However, FPO households also had greater consumption of unhealthy foods such as sweets and fried snacks as compared to non-FPO households.

## Women's empowerment

Approximately half of women in FPO and non-FPO households were empowered (Table 4). Access to and decisions about agricultural credit and self-help group membership were particularly low among both FPO and non-FPO women. There was no significant association between FPO membership and empowerment: adjusted odds ratio (95% CI), 0.96 (0.72, 1.30) (Table 7 in S1 Text). This finding was generally consistent across reported caste groups (Table 8 in S1 Text).

## Availability of nutrient dense foods and food expenditures

Villages in the region were dependent on three bi-weekly markets that took place in two FPO villages and one non-FPO village for vegetables and fruits. The survey dates aligned with only

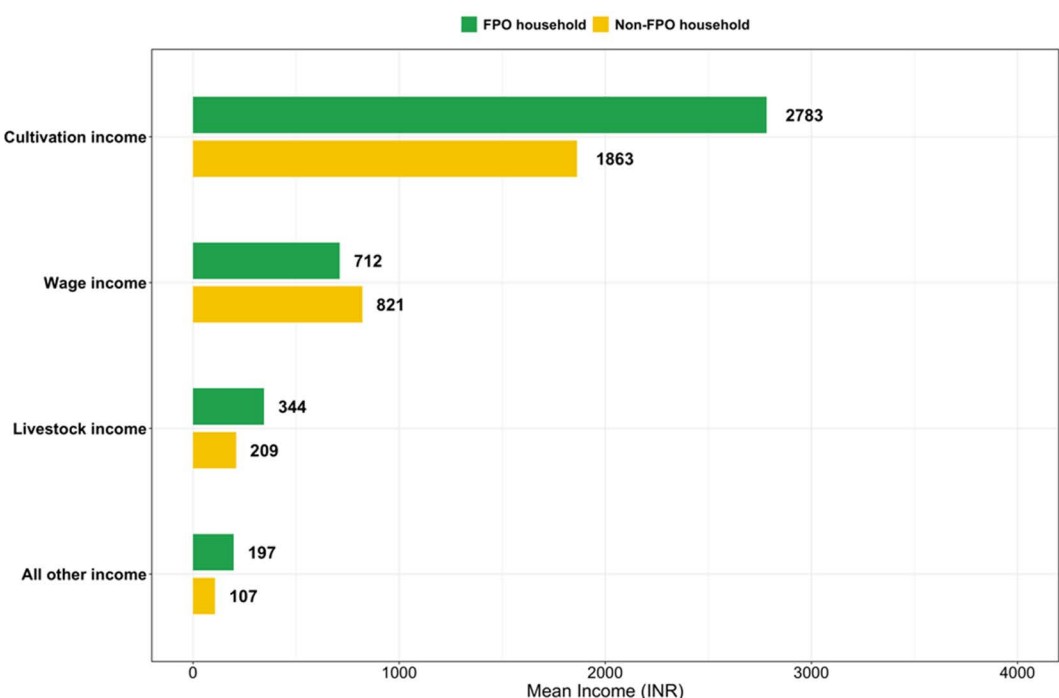

**Fig 2. Unadjusted mean monthly household income from cultivation, livestock, wages and other income of agricultural households participating in a cross-sectional survey comparing households with and without farmer producer organization (FPO) members in Uttar Pradesh, India (n = 819 households).**

**Table 4. Indicators of women's empowerment among women participating in a cross-sectional survey comparing households with and without farmer producer organization (FPO) members in Uttar Pradesh, India[1].**

| | FPO household<br>n = 412 | Non-FPO household<br>n = 388 |
|---|---|---|
| **Overall women's empowerment[2]** | 49% (201) | 53% (204) |
| **Women's empowerment in agriculture indicators[3]** | | |
| Input in productive decisions | 96% (396) | 94% (364) |
| Ownership of assets | 87% (358) | 88% (341) |
| Access to and decisions about credit | 3.2% (13) | 3.4% (13) |
| Control over use of income | 100% (412) | 98% (382) |
| Self-help group membership | 1% (4) | 0.5% (2) |
| Work balance | 49% (203) | 57% (220) |

[1]Values are % (n).

[2]Overall empowerment is achieved when a woman is empowered in at least three out of the five domains.

[3]Input in productive decisions: empowered if makes decisions or has input in decision in at least one area; Ownership of assets: empowered if solely or jointly owns at least one agricultural asset that is not a small agricultural asset; Access to and decisions about credit: empowered if solely or jointly make at least one decision regarding at least one source of credit; Control over use of income: empowered if has some input in how to use income and not only for minor household expenditures; Self-help group membership: empowered if actively participates in a self-help group; Work balance: empowered if time spent on productive tasks is less than 10.5 hours per day.

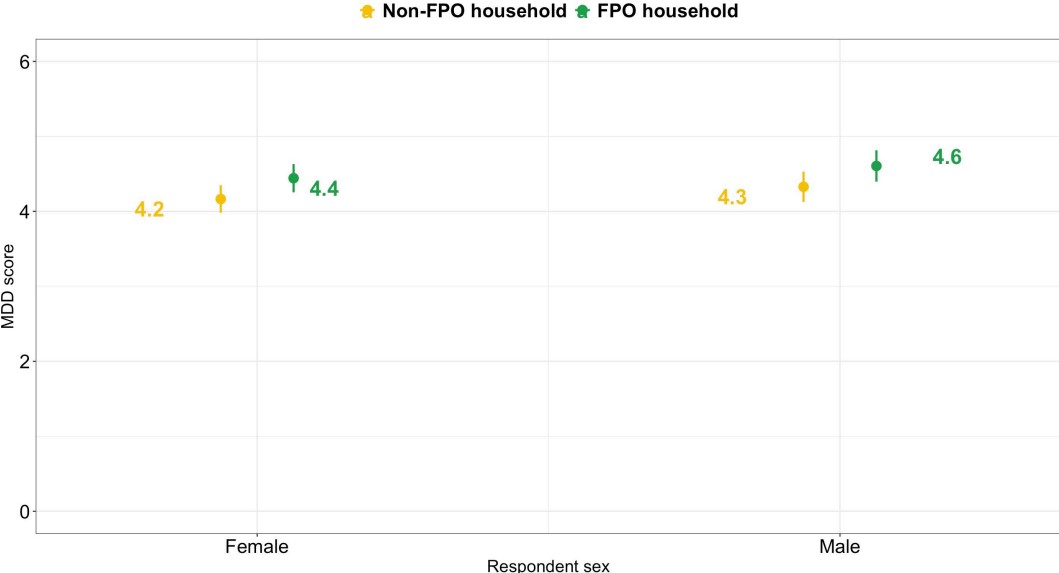

**Fig 3. Adjusted mean minimum dietary diversity score (95% confidence interval) among adult men and women participating in a cross-sectional survey comparing households with and without farmer producer organization (FPO) members in Uttar Pradesh, India.** Values are adjusted for educational attainment, caste, and total land cultivated in the kharif season (hectares).

one bi-weekly market. The density of grocery stores was similar between FPO and non-FPO villages (13 across 8 FPO villages and 9 across 5 non-FPO villages), as was the density of pan shops (10 across 8 FPO villages and 5 across 5 non-FPO villages). Millets were only available in 2 shops in FPO villages (not in any shops in non-FPO villages) but the availability of seeds and nuts was similar between FPO and non-FPO villages (13 shops sold them in FPO villages and 10 in non-FPO villages). Food expenditures per week were higher for FPO households than for non-FPO households: mean (SD) 1,816 INR (1,294) versus 1,624 INR (1,939).

## FPO support and challenges

Twenty-eight percent of FPO households reported receiving farm-related advisories in the past year. Advice was largely from the FPO (89% versus 8% from the government or KVK) and related to weather (92%), pests (89%), and market prices (84%). Nineteen percent of FPO households reported receiving training on crop production in the past year, the vast majority (97%) from the FPO itself (Table 9 in S1 Text).

Only 10% of FPO households purchased any inputs from the FPO in the past year, mostly pesticides (80% of those who purchased any), fertilizer and seeds (both 65%). Only 1 FPO household used any of the processing facilities of the FPO in the past year on a rental basis (a daal mill). The three qualitative interviews suggested a need for a designated platform to facilitate the exchange of knowledge and skills between government stakeholders and FPO members. Such a platform could be used to address barriers such as poor access to credit and could provide networks for forward market linkages to increase the price FPO members receive for their produce.

## Discussion

FPOs are a potentially effective strategy for improving crop diversity, farmers' incomes, and diet diversity in India, though more research is needed before causal claims can be made.

As this was a cross-sectional study, we cannot rule out reverse causality – it is possible that farmers with greater incomes are more likely to join FPOs. The findings of our pilot support the need for more rigorous assessments such as randomized controlled trials. Moreover, we found some evidence that the greater income has mixed effects on the healthfulness of household diets, with higher intakes of both healthy foods such as vegetables, fruits, dairy, and fish, as well as unhealthy foods such as sweets and fried foods among adults in FPO households. The findings of this study can be used to inform evidence-based policies to provide dedicated support for FPOs and improve policy convergence between rural development, agriculture, and nutrition. Such policies should consider integrating FPO support measures such as post-harvest technologies to introduce affordable nutrient-rich foods for year-round sale in rural markets, as well as implementing a national tax on high fat, sugar, and salt products given evidence that health taxes represent an effective win-win for public health and public finances [34–36]. Two important gaps for further research include (1) a deeper understanding of the challenges faced by FPOs, including market dynamics as well as barriers faced by the farmers in availing of FPO services, and (2) testing which specific services delivered to FPO members are most effective for improving outcomes such as income, nutrition, empowerment, and sustainability. FPOs can provide a number of services, including facilitating access to credit, providing storage and processing facilities, reducing the cost of inputs (through economies of scale), increasing bargaining power, opening doors to new markets, and/or delivering training relating to agricultural practices, new technologies, value-add activities, business, management, etc. [37,38]. Evidence suggests it may be better to focus on providing a few services well [38], which would require further research to understand which services would be most effective for which outcomes across different contexts.

While we found FPO households cultivated a greater number of crops that was statistically significant, the effect size was small (adjusted mean difference of 0.30 crops) and the most common crops grown were staples and cash crops (e.g., rice, wheat, red chili, and sugarcane). This effect size is comparable to, for example, a recent program implemented by Bangladesh's Ministry of Agriculture which had the explicit aim of increasing production diversity (effect size of 0.3–0.4 crops) [39], but evidence from previous studies suggests that much larger increases in on-farm crop diversity are needed to impact nutritional outcomes such as diet diversity [40]. However, smaller increases in on-farm crop diversity—for example, comparing one crop (monocropping) to more than one crop—may be associated with enhanced biodiversity [41]. The impact of FPO membership on crop diversity remains largely unexplored in the literature. The few existing studies focused on crop cultivars rather than crop species [7]. Many factors influence farmers' decisions to grow certain crops, including market access, agroclimatic conditions, farmers' knowledge and experience, and access to inputs, among other factors [42]. Further work is needed to understand whether and how FPOs can be a lever for improving crop diversity in the Indian context.

Many studies have demonstrated that membership in farmer cooperatives is associated with an increase in household income [2,3], including cross-sectional studies comparing FPO and non-FPO households in Bihar and Gujarat [17,19]. Likewise, we found that FPO households had a higher household income than non-FPO households. However, our study was cross-sectional, and it is possible that households with higher incomes are more likely to join FPOs. Prospective research is needed to confirm the direction of this relationship. The latest Situation Assessment Survey of Agricultural Households in Uttar Pradesh (July 2018-June 2019) found that the average monthly income for agricultural households was Rs.8,061 (p. 211 of [24]), which is more than twice the income reported by households in our sample (~Rs.4,035 per month for FPO households and ~ Rs.3,000 per month for non-FPO households). Farmers in our sample were making less from wages than farmers in this previous

survey which could partially explain this result. The context of our study, Fatehpur district, also has a lower economic status than the state of Uttar Pradesh, on average. For example, according to the latest National Family Health Survey (2019–2021), only 77% of the population in Fatehpur is living in households with electricity, 63% use an improved sanitation facility, and 39% use clean fuel for cooking compared to 91%, 69%, and 50% of households, respectively, at the state level [23]. Female literacy rates in the district are also lower than the state average: 62% versus 66%, respectively [23]. These factors could also partially explain the lower income observed in our sample as compared to the state average. Other explanations could be different survey instruments used or under-reporting by participants, among other factors. For these reasons, the absolute values of income should not be overly interpreted.

We explored whether the higher cultivation income could be due to higher yields or higher sales prices in this sample of farmers. FPO households in our sample had slightly higher yields of paddy and wheat than non-FPO households. The literature is mixed on whether cooperative membership has a significant effect on yields [6]. A study in Ethiopia found that cooperative membership increased the market price and quantity sold by women honey farmers [9]. We found that most households—both FPO and non-FPO—in this sample did not sell their paddy or wheat. It is possible that some farmers who were planning to sell their produce had not yet sold it, as data were collected between 28 April and 30 May. Among those who did sell their produce, which was less than half of the sample and so should be cautiously interpreted, FPO households sold slightly greater quantities, but received slightly lower prices than non-FPO households—possibly because they sold to traders more frequently than non-FPO households who were more likely to sell to village markets. However, the differences were small (~Rs.163/Quintal for paddy and ~ Rs.119/Quintal for wheat) and could reflect reporting errors rather than true differences in sale price. The proportion of both FPO and non-FPO households selling at government markets was relatively low, especially for paddy (only 3.4% of FPO households and 6.5% of non-FPO households sold paddy to government markets), and on average, households received a price less than or only slightly more than the minimum support price (MSP) for paddy and wheat [43]: MSP Rs.2,183 for common paddy in 2023/24 versus mean sale price of Rs.1,491 for FPO households and Rs.1,654 for non-FPO households and MSP Rs.2,125 for wheat in 2023/24 versus mean sale price of Rs.2,022 for FPO households and Rs.2,142 for non-FPO households. Future research should focus on investigating these market dynamics for FPO versus non-FPO households.

Adult men and women in FPO households were more likely to have diverse diets, driven by greater consumption of vegetables, fruits, dairy, and fish. The pathway to year-round diverse diets in this context is likely to be primarily through higher incomes, in combination with improved market access and, to a lesser extent, greater crop diversity. Previous studies in India have found that improving market access and income diversification are critical pathways for improving diet diversity in agricultural households, and mixed effects for the importance of crop diversification, depending on the context [44,45]. A recent scoping review of welfare impacts of farmer cooperatives did not identify any previous studies that explored the impact of membership on diet diversity in more than 200 studies identified [38]. Our findings suggest improved diet diversity may be a promising co-benefit of FPO membership. However, the higher consumption of unhealthy foods such as sweets and fried foods that was also observed among FPO members suggests the need for greater policy convergence between agriculture and nutrition agendas. Previous studies suggest that nutrition behavior change communication (BCC) to groups such as self-help groups in India has a limited effect on nutritional status, particularly when it is an under-resourced add-on intervention and does not have a sustained high intensity of implementation [46–50]. Moreover, nutrition-sensitive agriculture programs with a BCC component tend to focus on undernutrition and promoting

healthy foods, rather than discouraging unhealthy foods (often, consumption of unhealthy foods is not even assessed) [46,48–50]. Thus, policies should instead consider changes to the food environment such as the introduction of affordable nutrient-rich foods for year-round sale in rural markets and a national tax on high fat, sugar, and salt products [34–36].

We did not find that women who were members of FPOs were more empowered than non-FPO women, in terms of input in agricultural production decisions, ownership of assets, access to and decisions about credit, or control over income. These findings contrast to those from a study in Uganda which found that being part of a cooperative increased women's negotiating skills and decision-making power [8]. In particular, access to and decisions about credit seems to be low amongst women in Uttar Pradesh and could be a key area for improvement given evidence from a previous study in Bihar that lack of credit is the primary barrier preventing the adoption of new technologies by FPO members [17]. Several reasons have been proposed for low credit access among women in rural India including inactive accounts [51] and low engagement (financial institutions often give credit based on an applicant's transaction history); lack of collateral due to limited access to assets and property; distance to financial institutions; and lack of women in frontline roles in the financial sector [52,53]. Financial and digital literacy programs for women, enabling the use of mobile technology to access accounts and credit, gender-sensitive innovations in credit scoring for women farmers [54], and increasing access to collateral-free loans could help address these barriers.

This study is not without limitations. Surveys were conducted at one time point. It may be the case that the FPO households already had more crop production diversity, higher incomes, greater diet diversity, and women's empowerment before becoming members of the FPO. While we adjusted for educational attainment, caste, and farm size, FPO and non-FPO households may have differed with regards to other characteristics not captured by the survey, and those differences may also explain differences observed in the outcomes. For example, previous studies have found that access to Internet, primary occupation in farming, years of farming experience, contact with agricultural extension workers or receipt of information from government agricultural agencies, and farmers' intentions (e.g., to expand operations) may influence FPO membership [17–19,21]. A randomized controlled trial would be the gold standard approach to determining the impact of FPO membership and could be considered in future work. Another limitation is that we only evaluated one FPO and studies have shown heterogenous impacts across FPOs with varying memberships [38]. Moreover, we used quota sampling rather than random sampling, which may have introduced sampling bias. Thus, our findings may not be generalizable to all FPOs in Uttar Pradesh or elsewhere in India. Finally, we were not able to conduct subgroup analyses by landholding size given the small number of medium and large farms in our sample (96% of our sample was small farms). Future studies should evaluate whether impacts vary across small, medium, and large farms given evidence from previous studies that farms of all sizes may not benefit equally from membership in cooperatives [2,3].

In order to achieve the stated goals and other co-benefits of FPOs, the emphasis should not just be on the quantity of FPOs but also the quality of services provided. Such services could include input purchasing, agricultural machinery leasing, value-addition and processing (product differentiation), marketing, market linkages including for export, more reliable contracts, and finance, among others. However, a single FPO cannot be effective at providing all of these services. As a next step, it will be important to understand what services are most important and have the biggest impact on farmers' income, diet quality, and women's empowerment in a given context. Collection of information regarding the services provided by registered FPOs should be considered by the Ministry of Corporate Affairs and other FPO portals as such data are currently lacking and are essential for developing effective policies to improve FPO performance [15].

FPO membership may contribute to greater crop diversity, higher incomes from cultivation, and greater diet diversity among farming households in Fatehpur district of Uttar Pradesh, India. However, the high consumption of unhealthy foods observed in this survey suggest that work is needed to encourage expenditure on nutrient-rich foods. Moreover, access to and decisions about credit should be the focus of future programs aiming to increase women's empowerment in this context. Together, these actions may accelerate progress towards achieving several UN Sustainable Development Goals in India, including especially no poverty (#1), zero hunger (#2), good health and well-being (#3), gender equality (#5), and decent work and economic growth (#8).

## Supporting Information

**S1 Checklist.  Inclusivity in global research.**
(DOCX)

**S1 Text.  Additional results tables.**
(DOCX)

**S1 File.  Household survey.**
(DOCX)

## Acknowledgments

We would like to thank the participants for kindly volunteering their time for surveys and interviews. We would also like to acknowledge the support of Ernst & Young LLP, working as a Technical Support Unit with the Agriculture Department in Uttar Pradesh, in facilitating our field study and organising stakeholder interviews at the village and state level.

## Author contributions

**Conceptualization:** Lindsay M. Jaacks, Divya Veluguri, Ananya Awasthi.

**Data curation:** Lindsay M. Jaacks, Nishmeet Singh, Kaela Connors.

**Formal analysis:** Lindsay M. Jaacks, Nishmeet Singh, Kaela Connors.

**Funding acquisition:** Ananya Awasthi.

**Investigation:** Lindsay M. Jaacks, Nishmeet Singh, Divya Veluguri, Kaela Connors, Aleesha Sooraj, Apoorva Kalra.

**Methodology:** Lindsay M. Jaacks, Nishmeet Singh, Divya Veluguri, Kaela Connors.

**Project administration:** Lindsay M. Jaacks, Divya Veluguri, Aleesha Sooraj.

**Supervision:** Lindsay M. Jaacks, Divya Veluguri, Apoorva Kalra, Ananya Awasthi.

**Validation:** Lindsay M. Jaacks, Kaela Connors.

**Visualization:** Lindsay M. Jaacks, Nishmeet Singh, Kaela Connors.

**Writing – original draft:** Lindsay M. Jaacks.

**Writing – review & editing:** Nishmeet Singh, Divya Veluguri, Kaela Connors, Aleesha Sooraj, Apoorva Kalra, Ananya Awasthi.

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
