## [Decision Letter · Decision Letter 0]

28 Nov 2024

PONE-D-24-46385Association of membership in a farmer producer organization with crop diversity, household income, diet diversity, and women’s empowerment: A mixed methods study in Uttar Pradesh, IndiaPLOS ONE

Dear Dr. Jaacks,

Thank you for submitting your manuscript to PLOS ONE. After careful consideration, we feel that it has merit but does not fully meet PLOS ONE’s publication criteria as it currently stands. Therefore, we invite you to submit a revised version of the manuscript that addresses the points raised during the review process.

We look forward to receiving your revised manuscript.

Kind regards,

Ananth JP

Academic Editor

PLOS ONE

Journal Requirements:

3. In the ethics statement in the Methods, you have specified that verbal consent was obtained. Please provide additional details regarding how this consent was documented and witnessed, and state whether this was approved by the IRB.

4. In the online submission form, you indicated that your data will be submitted to a repository upon acceptance. We strongly recommend all authors deposit their data before acceptance, as the process can be lengthy and hold up publication timelines. Please note that, though access restrictions are acceptable now, your entire minimal dataset will need to be made freely accessible if your manuscript is accepted for publication. This policy applies to all data except where public deposition would breach compliance with the protocol approved by your research ethics board. If you are unable to adhere to our open data policy, please kindly revise your statement to explain your reasoning and we will seek the editor's input on an exemption.

6. Please include captions for your Supporting Information files at the end of your manuscript, and update any in-text citations to match accordingly. Please see our Supporting Information guidelines for more information: http://journals.plos.org/plosone/s/supporting-information .

Reviewers' comments:

Reviewer's Responses to Questions

**Comments to the Author**

1. Is the manuscript technically sound, and do the data support the conclusions?

Reviewer #1: Yes

Reviewer #2: No

Reviewer #3: Partly

2. Has the statistical analysis been performed appropriately and rigorously? 

Reviewer #1: Yes

Reviewer #2: No

Reviewer #3: Yes

3. Have the authors made all data underlying the findings in their manuscript fully available?

Reviewer #1: Yes

Reviewer #2: No

Reviewer #3: Yes

4. Is the manuscript presented in an intelligible fashion and written in standard English?

Reviewer #1: Yes

Reviewer #2: Yes

Reviewer #3: Yes

5. Review Comments to the Author

Reviewer #1: This well-designed mixed-methods study provides valuable insights into FPO impacts on agricultural households in Uttar Pradesh. The research demonstrates robust methodology and statistical analysis, though causal relationships cannot be established. Findings have important implications for agricultural and nutrition policy.

Comments:

1. The study reports that "FPO households had slightly greater crop production diversity than non-FPO households (mean of 4 crops versus 3.5 crops, respectively)." How meaningful is this small difference in practical terms for agricultural sustainability and household nutrition?

2. The observed pattern of "lower sale prices for FPO households selling to traders" versus "higher prices for non-FPO households selling to village markets" appears counterintuitive given that FPOs are meant to improve market access. This warrants deeper investigation into market dynamics.

3. "While we adjusted for educational attainment, caste, and farm size, FPO and non-FPO households may have differed with regards to other characteristics not captured by the survey" - Could you identify specific unmeasured variables that might confound the relationships observed?

4. The study indicates mixed effects on diet quality, with increased consumption of both healthy and unhealthy foods in FPO households. What specific nutrition education interventions could be integrated into FPO programs to promote healthier dietary choices?

5. "Only 10% of FPO households purchased any inputs from the FPO in the past year" suggests very low utilization of FPO services. What are the barriers preventing members from taking advantage of these resources?

6. The statement that "28% of FPO households reported receiving farm-related advisories in the past year" seems quite low given that extension services are a key benefit of FPO membership. How could advisory outreach be improved?

7. The research notes income discrepancies: "The latest Situation Assessment Survey...found average monthly income was 8,061 INR... more than twice the income reported by households in our sample." Does this suggest potential methodological issues in income measurement?

8. While the study employed mixed methods, the qualitative component involving only three stakeholder interviews seems limited. Could expanding qualitative research provide deeper insights into the challenges and opportunities?

9. Given the finding that "access to and decisions about agricultural credit and self-help group membership were particularly low among both FPO and non-FPO women," what specific policy interventions could improve women's access to financial resources?

10. The study could be improved by integrating insights from recent research in related fields: https://doi.org/10.3390/ma15207098; https://doi.org/10.3390/ma15186222;

Reviewer #3: Overall comments: Manuscript PONE-D-24-46385 presents famer producer organizations (FPOs), a kind of member farmer cooperatives’ socio-economic impacts on income, crop diversity, and household livelihoods of the local farmers of a state of India. Research on membership of FPOs and their socio-economic impacts on the farmers in developing countries substantially reported in the literature and the manuscript does not sufficiently differentiate novel research aspects from existing literature for a clear articulation of the current findings’ contribution towards significantly advancement of the existing knowledge. The two outcomes of the membership of FPOs viz., women’s empowerment and dietary diversity adds bit of novelty, however; there is lack of in-depth exploration of the mechanisms through which FPO membership influences dietary patterns and gender dynamics of the women diversity in FPO, which put a limitation to the innovative contribution of the given study to nutrition-sensitive agriculture. Limited critical analyses of results on women’s empowerment which show no significant differences between FPO and non-FPO households. The study is based on limited sample size i.e., only a single FPO which put a limitation to generate broadly applicable insights and uncover unique patterns related to comparative analyses across different FPOs, regions, or organizational structures. The causal inferences related to FPO’s emerging benefits and impacts over time are extremely limited as the given study is cross-sectional in nature. The distinct theoretical and methodological framework lacking e.g., combining of the standard tools like the FAO minimum diet diversity score and A-WEAI index for generating a deeper unexpected insight of the FPO’s dynamics. The policy recommendation implications are general and are like the recommendations given in multiple studies. The detailed comments are given in the attached file.

6. PLOS authors have the option to publish the peer review history of their article (what does this mean? ). If published, this will include your full peer review and any attached files.

**Do you want your identity to be public for this peer review?** For information about this choice, including consent withdrawal, please see our Privacy Policy .

Reviewer #1: No

Reviewer #2: No

Reviewer #3: **Yes: ** Muhammad Shafiq

---

## [Author Response · Author response to Decision Letter 1]

24 Jan 2025

We thank the reviewers for their time and careful consideration of our manuscript. We have incorporated their suggestions, which we believe have improved the manuscript. We provide point-by-point responses below highlighting changes made in the text. Line numbers below refer to the clean version of the manuscript (without tracked changes).

Reviewer #1:

Overall comments: This well-designed mixed-methods study provides valuable insights into FPO impacts on agricultural households in Uttar Pradesh. The research demonstrates robust methodology and statistical analysis, though causal relationships cannot be established. Findings have important implications for agricultural and nutrition policy.

Response: Thank you for taking the time to review our manuscript and for recognizing the importance of this study. We appreciate your feedback, to which we have responded inline below.

1. The study reports that "FPO households had slightly greater crop production diversity than non-FPO households (mean of 4 crops versus 3.5 crops, respectively)." How meaningful is this small difference in practical terms for agricultural sustainability and household nutrition?

Response: We agree with the reviewer that this difference, while statistically significant (p=0.005), is not likely to be nutritionally significant. We have prefaced all statements regarding this difference with “slightly” and have expanded our discussion on the implications of a difference of this magnitude in the revised manuscript. (p. 21, lines 377-383)

“This effect size is comparable to, for example, a recent program implemented by Bangladesh’s Ministry of Agriculture which had the explicit aim of increasing production diversity (effect size of 0.3-0.4 crops) [39], but evidence from previous studies suggests that much larger increases in on-farm crop diversity are needed to impact nutritional outcomes such as diet diversity [40]. However, smaller increases in on-farm crop diversity—for example, comparing one crop (monocropping) to more than one crop—may be associated with enhanced biodiversity [41].”

2. The observed pattern of "lower sale prices for FPO households selling to traders" versus "higher prices for non-FPO households selling to village markets" appears counterintuitive given that FPOs are meant to improve market access. This warrants deeper investigation into market dynamics.

Response: We agree with the reviewer that this warrants deeper investigation into market dynamics in future studies and have added this point to the revised discussion. However, we caution against over-interpretation or generalization of these findings from our analysis given that less than half of households in our sample sold any of their produce and a very small percentage (<4%) had fully sold their produce (Table 2). The sale price is therefore only available for a small percentage of the sample (17% of FPO and non-FPO households cultivating wheat, 9% of FPO households cultivating paddy, and 7% of non-FPO households cultivating paddy). For this reason, we did not conduct any formal statistical testing of differences between FPO and non-FPO households for this exploratory analysis. The mean differences are small (~163 INR/quintal for paddy and ~119 INR/quintal for wheat compared to SDs of 449-525 INR/quintal). We therefore now preface all statements regarding this difference with “slightly”. (pp. 23-24, lines 415-431)

“We found that most households—both FPO and non-FPO—in this sample did not sell their paddy or wheat. It is possible that some farmers who were planning to sell their produce had not yet sold it, as data were collected between 28 April and 30 May. Among those who did sell their produce, which was less than half of the sample and so should be cautiously interpreted, FPO households sold slightly greater quantities, but received slightly lower prices than non-FPO households—possibly because they sold to traders more frequently than non-FPO households who were more likely to sell to village markets. However, the differences were small (~163 INR/quintal for paddy and ~119 INR/quintal for wheat) and could reflect reporting errors rather than true differences in sale price. The proportion of both FPO and non-FPO households selling at government markets was relatively low, especially for paddy (only 3.4% of FPO households and 6.5% of non-FPO households sold paddy to government markets), and on average, households received a price less than or only slightly more than the minimum support price (MSP) for paddy and wheat [43]: MSP 2,183 for common paddy in 2023/24 versus mean sale price of 1,491 for FPO households and 1,654 for non-FPO households and MSP 2,125 for wheat in 2023/24 versus mean sale price of 2,022 for FPO households and 2,142 for non-FPO households. Future research should focus on investigating these market dynamics for FPO versus non-FPO households.”

3. "While we adjusted for educational attainment, caste, and farm size, FPO and non-FPO households may have differed with regards to other characteristics not captured by the survey" - Could you identify specific unmeasured variables that might confound the relationships observed?

Response: Thank you for this suggestion. We have listed examples of other characteristics not captured by the survey that could be considered in future studies to the revised discussion. (pp. 25-26, lines 473-477)

“For example, previous studies have found that access to Internet, primary occupation in farming, years of farming experience, contact with agricultural extension workers or receipt of information from government agricultural agencies, and farmers’ intentions (e.g., to expand operations) may influence FPO membership [17–19,21].”

4. The study indicates mixed effects on diet quality, with increased consumption of both healthy and unhealthy foods in FPO households. What specific nutrition education interventions could be integrated into FPO programs to promote healthier dietary choices?

Response: Thank you for suggesting this. We have expanded our discussion of evidence-based policy options to promote healthier dietary choices in the revised manuscript. (pp. 20-21 lines 358-362 & p. 24 lines 444-452)

“Such policies should consider integrating FPO support measures such as postharvest technologies to introduce affordable nutrient-rich foods for year-round sale in rural markets, as well as implementing a national tax on high fat, sugar, and salt products given evidence that health taxes represent an effective win-win for public health and public finances [34–36].”

“Previous studies suggest that nutrition behavior change communication (BCC) to groups such as self-help groups in India has a limited effect on nutritional status, particularly when it is an under-resourced add-on intervention and does not have a sustained high intensity of implementation [46–50]. Moreover, nutrition-sensitive agriculture programs with a BCC component tend to focus on undernutrition and promoting healthy foods, rather than discouraging unhealthy foods (often, consumption of unhealthy foods is not even assessed) [46,48–50]. Thus, policies should instead consider changes to the food environment such as the introduction of affordable nutrient-rich foods for year-round sale in rural markets and a national tax on high fat, sugar, and salt products [34–36].”

5. "Only 10% of FPO households purchased any inputs from the FPO in the past year" suggests very low utilization of FPO services. What are the barriers preventing members from taking advantage of these resources?

Response: As this study was an impact evaluation, unfortunately, we did not collect process evaluation information such as specific barriers faced by individual FPO members. We have highlighted this as a key gap to be addressed by future research in the revised discussion. (p. 21, lines 362-373)

“Two important gaps for further research include (1) a deeper understanding of the challenges faced by FPOs, including market dynamics as well as barriers faced by the farmers in availing of FPO services, and (2) testing which specific services delivered to FPO members are most effective for improving outcomes such as income, nutrition, empowerment, and sustainability. FPOs can provide a number of services, including facilitating access to credit, providing storage and processing facilities, reducing the cost of inputs (through economies of scale), increasing bargaining power, opening doors to new markets, and/or delivering training relating to agricultural practices, new technologies, value-add activities, business, management, etc. [37,38]. Evidence suggests it may be better to focus on providing a few services well [38], which would require further research to understand which services would be most effective for which outcomes across different contexts.”

6. The statement that "28% of FPO households reported receiving farm-related advisories in the past year" seems quite low given that extension services are a key benefit of FPO membership. How could advisory outreach be improved?

Response: As this study was an impact evaluation, unfortunately, we did not collect process evaluation information such as specific improvements that could be made to increase access and utilization of FPO services. We have highlighted this as a key gap to be addressed by future research in the revised discussion (please see response to comment #5 for new text added to the revised manuscript).

7. The research notes income discrepancies: "The latest Situation Assessment Survey...found average monthly income was 8,061 INR... more than twice the income reported by households in our sample." Does this suggest potential methodological issues in income measurement?

Response: The survey instrument used to assess household income in our study was adapted from the survey instrument used in the latest Situation Assessment Survey of Agricultural Households. Nonetheless, as the reviewer points out, it is possible that there was misreporting, particularly under-reporting of income in our study. However, it is more likely that the income discrepancy stems from the fact that our sample population is poorer than the state average. We have clarified this, citing data from the latest round of the National Family Health Survey (2019-2021), in the revised discussion. (pp. 22-23, lines 400-407)

“The context of our study, Fatehpur district, also has a lower economic status than the state of Uttar Pradesh, on average. For example, according to the latest National Family Health Survey (2019-2021), only 77% of the population in Fatehpur is living in households with electricity, 63% use an improved sanitation facility, and 39% use clean fuel for cooking compared to 91%, 69%, and 50% of households, respectively, at the state level [23]. Female literacy rates in the district are also lower than the state average: 62% versus 66%, respectively [23]. These factors could also partially explain the lower income observed in our sample as compared to the state average.”

8. While the study employed mixed methods, the qualitative component involving only three stakeholder interviews seems limited. Could expanding qualitative research provide deeper insights into the challenges and opportunities?

Response: Thank you for this suggestion. The “mixed methods” aspect of this study included the household survey, the market basket survey (direct observation of all open and operating vendors in 13 villages) as well as the semi-structured qualitative interviews with key stakeholders (the CEO of the FPO, a representative of the state agriculture department, and a district-level government administrative official). While we agree with the reviewer that further qualitative interviews may provide additional insights into the challenges and opportunities for FPOs, this was outside the scope of the current study—these interviews were conducted in order to provide further context for interpreting the results of the household surveys. As one of the first studies in India to empirically evaluate the impact of FPOs, we feel this project will open doors for further qualitative research as a next step, and we highlight this point in the revised discussion. (p. 21, lines 362-373)

“Two important gaps for further research include (1) a deeper understanding of the challenges faced by FPOs, including market dynamics as well as barriers faced by the farmers in availing of FPO services, and (2) testing which specific services delivered to FPO members are most effective for improving outcomes such as income, nutrition, empowerment, and sustainability.”

9. Given the finding that "access to and decisions about agricultural credit and self-help group membership were particularly low among both FPO and non-FPO women," what specific policy interventions could improve women's access to financial resources?

Response: We have added further detail to the revised discussion regarding what specific interventions are available to improve women’s access to credit in the Indian context. (p. 25, lines 460-467)

“Several reasons have been proposed for low credit access among women in rural India including inactive accounts [51] and low engagement (financial institutions often give credit based on an applicant’s transaction history); lack of collateral due to limited access to assets and property; distance to financial institutions; and lack of women in frontline roles in the financial sector [52,53]. Financial and digital literacy programs for women, enabling the use of mobile technology to access accounts and credit, gender-sensitive innovations in credit scoring for women farmers [54], and increasing access to collateral-free loans could help address these barriers.”

10. The study could be improved by integrating insights from recent research in related fields: https://doi.org/10.3390/ma15207098; https://doi.org/10.3390/ma15186222.

Response: We think there may have been an error in the DOIs provided as these refer to studies on materials science (“Fly Ash-Based Geopolymer Composites: A Review of the Compressive Strength and Microstructure Analysis” and “Concrete Containing Waste Glass as an Environmentally Friendly Aggregate: A Review on Fresh and Mechanical Characteristics”), which are not relevant to our study on farmer producer organizations in India.

Reviewer #3:

Overall comments: Manuscript PONE-D-24-46385 presents famer producer organizations (FPOs), a kind of member farmer cooperatives’ socio-economic impacts on income, crop diversity, and household livelihoods of the local farmers of a state of India. Research on membership of FPOs and their socio-economic impacts on the farmers in developing countries substantially reported in the literature and the manuscript does not sufficiently differentiate novel research aspects from existing literature for a clear articulation of the current findings’ contribution towards significantly advancement of the existing knowledge. The two outcomes of the membership of FPOs viz., women’s empowerment and dietary diversity adds bit of novelty, however; there is lack of in-depth exploration of the mechanisms through which FPO membership influences dietary patterns and gender dynamics of the women diversity in FPO, which put a limitation to the innovative contribution of the given study to nutrition-sensitive agriculture. Limited critical analyses of results on women’s empowerment which show no significant differences between FPO and non-FPO households. The study is based on limited sample size i.e., only a single FPO which put a limitation to generate broadly applicable insights and uncover unique patterns related to comparative analyses across different FPOs, regions, or organizational structures. The causal inferences related to FPO’s emerging benefits and impacts over time are extremely limited as the given study is cross-sectional in nature. The distinct theoretical and methodological framework lacking e.g., combining of the standard tools like the FAO minimum diet diversity score and A-WEAI index for generating a deeper unexpected insight of the FPO’s dynamics. The policy recommendation implications are general and are like the recommendations given in multiple studies. The detailed comments are as un

---

## [Editor Report · Decision Letter 1]

6 Feb 2025

Association of membership in a farmer producer organization with crop diversity, household income, diet diversity, and women’s empowerment in Uttar Pradesh, India

PONE-D-24-46385R1

Dear Dr. Jaacks,

We’re pleased to inform you that your manuscript has been judged scientifically suitable for publication and will be formally accepted for publication once it meets all outstanding technical requirements.

Kind regards,

Ananth JP

Academic Editor

PLOS ONE
---

## [Editor Report · Acceptance letter]

PONE-D-24-46385R1

PLOS ONE

Dear Dr. Jaacks,

I'm pleased to inform you that your manuscript has been deemed suitable for publication in PLOS ONE. Congratulations! Your manuscript is now being handed over to our production team.

Kind regards,

on behalf of

Dr. Ananth JP

Academic Editor

PLOS ONE